

# Evolutionary persistence in *Gunnera* and the contribution of southern plant groups to the tropical Andes biodiversity hotspot

Christine D. Bacon[1,2,3], Francisco J. Velásquez-Puentes[3,4], Luis Felipe Hinojosa[5], Thomas Schwartz[1], Bengt Oxelman[1], Bernard Pfeil[1], Mary T.K. Arroyo[5], Livia Wanntorp[6] and Alexandre Antonelli[1,2,7,8]

[1] Department of Biological and Environmental Sciences, University of Gothenburg, Gothenburg, Sweden
[2] Gothenburg Global Biodiversity Centre, Gothenburg, Sweden
[3] Laboratório de Biología Molecular (CINBIN), Universidad Industrial de Santander, Bucaramanga, Colombia
[4] Departamento de Química y Biología, Universidad del Norte, Barranquilla, Colombia
[5] Institute of Ecology and Biodiversity, Facultad de Ciencias, Universidad de Chile, Santiago, Chile
[6] Department of Phanerogamic Botany, Swedish Museum for Natural History, Stockholm, Sweden
[7] Gothenburg Botanical Garden, Gothenburg, Sweden
[8] Department of Organismic and Evolutionary Biology, Harvard University, Cambridge, MA, USA

Corresponding author
Christine D. Bacon,
christine.bacon@bioenv.gu.se,
christinedbacon@gmail.com

## ABSTRACT

Several studies have demonstrated the contribution of northern immigrants to the flora of the tropical Andes—the world's richest and most diverse biodiversity hotspot. However, much less is known about the biogeographic history and diversification of Andean groups with southern origins, although it has been suggested that northern and southern groups have contributed roughly equally to the high Andean (i.e., páramo) flora. Here we infer the evolutionary history of the southern hemisphere plant genus *Gunnera*, a lineage with a rich fossil history and an important ecological role as an early colonising species characteristic of wet, montane environments. Our results show striking contrasts in species diversification, where some species may have persisted for some 90 million years, and whereas others date to less than 2 Ma since origination. The outstanding longevity of the group is likely linked to a high degree of niche conservatism across its highly disjunct range, whereby *Gunnera* tracks damp and boggy soils in cool habitats. Colonisation of the northern Andes is related to Quaternary climate change, with subsequent rapid diversification appearing to be driven by their ability to take advantage of environmental opportunities. This study demonstrates the composite origin of a mega-diverse biota.

## INTRODUCTION

Among the 34 biodiversity hotspots currently recognized, the tropical Andes is the richest and most diverse, comprising some 30,000 plant species (*Myers et al., 2000*). This equates to nearly a tenth of the world's flora contained in less than one per cent of its land area.

The tropical Andean hotspot, which stretches from western Venezuela to northern Chile and Argentina, constitutes an ideal arena for investigating the role of historical migrations in generating the exceptional plant species diversity found in the American tropics (the Neotropics).

Most evolutionary models proposed to explain Neotropical diversity (*Antonelli & Sanmartín, 2011b*; *Rull, 2011*) postulate a major role for *in situ* diversification, e.g., in Amazonia (e.g., *Haffer, 1969*; *Smith et al., 2014*, but see *Dexter et al., 2017*) and the Andes (*Gentry, 1982*; *Luebert & Weigend, 2014*). The relative contribution of immigrant lineages to modern Neotropical diversity is less understood, but has certainly played an important role (*Villagrán & Hinojosa, 1997*). For example, northern immigrants contributed more to the species diversity of the high elevation Andean páramo than southern immigrants (e.g., *Sklenář, Dušková & Balslev, 2011*). The contribution of immigrant taxa to modern Neotropical diversity may be primarily explained by either continuous range expansions from neighboring regions or long-distance dispersal, both from what today are temperate lineages into tropical latitudes (often facilitated by climatic change and mountain building), and from other trans-oceanic tropical regions (*Antonelli et al., 2015*). Modern distributions reflect ancestral ecological requirements (niche conservatism; *Wiens et al., 2010*), but they also are affected by biome shifts, such as adaptation of cool temperate immigrants into cold tropical areas (*Donoghue & Edwards, 2014*). More rarely, migration events can also be directly associated with physiological adaptations intro new environments (*Crisp et al., 2009*; *Simon et al., 2009*).

One of the characteristic elements of the Andes is the plant genus *Gunnera* (Gunneraceae; Fig. S1). Although eleven *Gunnera* species are reported in the páramo (*Luteyn, 1999*), most of these are found in montane forests and only one is a strict páramo species, *G. magellanica*—which is also found in the southern temperate Andes in wet habitats both below and above treeline (*Sklenář, Dušková & Balslev, 2011*). *Gunnera* has been present in montane forests since at least the Middle Pliocene in Colombia (*Hooghiemstra, Wijninga & Cleef, 2006*). In comparison to páramo species from southern regions, fewer montane forest species, such as *Gunnera*, have successfully colonised the northern Andes (*Wanntorp & Wanntorp, 2003*).

*Gunnera* includes 58 species primarily of the Southern Hemisphere (Africa, New Zealand, South America, and Tasmania), but also reaches Hawaii, Mexico, and Southeast Asia (Fig. 1; Table S1; *Bader, 1961*; *Mora-Osejo, Pabón-Mora & González, 2011*). Despite this wide geographical distribution encompassing all southern continents apart from Antarctica, the majority of extant species of *Gunnera* (up to 40 species in subgenus Panke) are distributed in Central and South America (*Mora-Osejo, Pabón-Mora & González, 2011*), most of them within the northern Andean biodiversity hotspot. However, *Gunnera* had an even wider geographic distribution during the Cretaceous, as demonstrated by numerous fossil pollen records from the Antarctic Peninsula, Australia, the Kerguelen Plateau, as well as in both North and South America (*Jarzen, 1980*). The oldest of these dates to the Turonian (ca. 90 Ma) of Peru (*Brenner, 1968*) and ten million years later *Gunnera* became widespread across all landmasses that previously formed Gondwana (*Jarzen, 1980*). Initial evidence suggested that biogeographic patterns in *Gunnera* are in

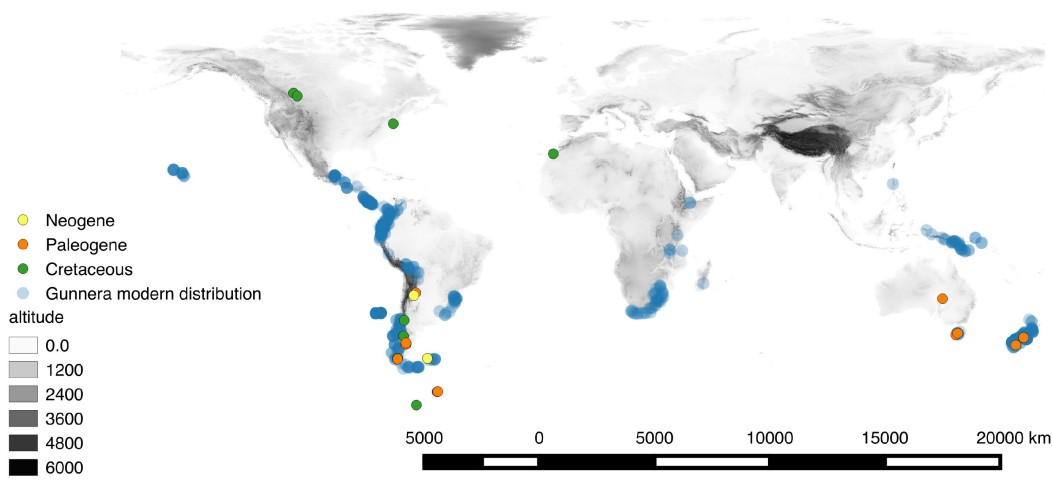

**Figure 1** **Map of the extant distribution of *Gunnera*, where high numbers of georeferences are reflected by darker blue colour.** The *Tricolpites reticulatus* pollen fossil was used to calibrate the *Gunnera* phylogeny in absolute time for this study and is also mapped through major geological time periods. Elevation is shown in grey scale where the lowest and highest global elevations are found in white and black, respectively.

agreement with the sequence of Gondwanan break-up, prompting *Wanntorp & Wanntorp (2003)* to suggest a Gondwanan origin for the genus and vicariance by continental drift as a plausible hypothesis to explain the present-day distribution of most species in the genus. However, these conclusions involved neither formal ancestral area reconstruction nor molecular dating. The study also did not explicitly investigate the biogeographical history of the Andean species comprising subgenus *Panke*.

The current widespread distribution of *Gunnera,* its rich fossil record and the many still unanswered questions regarding its biogeographic history all make *Gunnera* an ideal target for investigating the role of southern immigrants in the assembly of the flora of the Andean biodiversity hotspot. Here we infer the biogeographic, climatic, and evolutionary history of *Gunnera* to explicitly address the following questions: Where did *Gunnera* most likely originate? How and when did it attain its current distribution? When did it colonise the northern Andes where it is most diverse? Are areas of high diversity in the genus associated with higher rates of net diversification, or is diversity instead due to gradual accumulation of species? To what extent have species of *Gunnera* tracked the ancestral climatic niche? We also compare *Gunnera* with other Andean groups. Our study sheds further light on the geographical and temporal origins and composition of the highly diverse Andean flora.

## MATERIAL AND METHODS

### Phylogeny and divergence time estimation

Taxon sampling was complete at the species level for five of six subgenera of *Gunnera* (27 out of 58 species, ca. 46%). Within subgenus *Panke* we could only obtain material for 14 of the up to 40 species, because many species are only known from few collections or localities, hence the difficulty in defining the exact number of species (*Mora-Osejo, Pabón-Mora &*

*González, 2011*). New DNA sequence data was generated following the respective author protocols for the chloroplast regions *rps16* (*Oxelman, Liden & Berglund, 1997*) *rpoC1* and *ycf5* from the Consortium for the Barcode of Life (http://www.barcodeoflife.org), *psbA-trnH* (*Pang et al., 2012*) as well as the nuclear genes ITS (*Blattner, 1999*) and SEX4 (*Kotting et al., 2009*). All new sequences generated in this study have been deposited in GenBank (File S1). Nucleotide alignments were obtained independently for each of the loci using default parameters for MUSCLE in Geneious (Biomatters, Auckland, New Zealand). Due to poor alignability, the ITS sequences of *G. herteri* and *G. perpensa* were removed. We used the Akaike Information Criterion implemented in MrModelTest 2.2 (*Nylander, 2004*) to select the best-fitting nucleotide substitution model(s) and incorporated it in the inference of the species tree.

We used STACEY 1.04 (see http://www.indriid.com/software.html) in BEAST 2.3.0 and the DISSECT method (*Jones, Aydin & Oxelman, 2014*) to infer a multispecies coalescent tree. The method uses a version of the birth/death branching model for the species tree, which assigns high probabilities for branching events close to time zero; how close is defined by the "collapse height" parameter, which should be set as small as possible (see *Jones, Aydin & Oxelman, 2014*). The approach thus enables simultaneous exploration of species tree and species delimitation space. Individuals or groups of individuals known to belong to the same single species are operationally defined as minimal species. Clusters below the collapse height are considered to belong to the same species, as defined by the multispecies coalescent model. Here we defined all sequences from the same individual as minimal species. A lognormal (mean 4.6, standard deviation 2) growth rate prior distribution was used for the species tree. The growth rate is roughly 1 divided by the branch length, so that 95% of the distribution falls within the interval [2, 5,000] with median $e^{4.6} \approx 100$. Beta priors with shape parameters 1, 1 (resulting in uniform distributions) were used on collapse weight and relative death rates. A lognormal ($-7$, 2) prior was used for popPriorScale, which should approximate the average time to coalescence between two gene copies. In order to scale branch lengths in substitutions per site, the ITS rate was set to 1 and lognormal (0, 1) priors for the relative rates of the cpDNA and SEX4 trees were used. Collapse height was set to 0.0001. Ploidy was set to 1 for cpDNA and 2 for the two nuclear genes. The substitution model was GTR with a gamma prior distribution (0.05, 10) on each substitution type, with rate variation among sites was modeled with four rate categories for all three loci. Each locus also had branch rates constrained to an uncorrelated lognormal clock. The MCMC was run for 100 million generations and all parameters had effective sample sizes >180 after removing the first 10 million generations as burn-in. The maximum clade credibility species tree was generated by sampling trees every 50,000th generation in the stationary phase (the last 90 million generations), where the heights are common ancestor heights, scaled in substitutions/site.

A fossil *Gunnera* pollen grain was used to calibrate the phylogeny. *Tricolpites reticulatus* from the Turonian of Peru (*Brenner, 1968*) represents the first unambiguous appearance of the genus. Based on this calibration point, the Turonian/Coniacian boundary (Late Cretaceous) was used to provide a crown age of *Gunnera* by scaling the root of the STACEY tree (see above) using a mean age of 90 Ma. Its placement on the crown of *Gunnera* is

based on a morphological review of extant and fossil pollen of *Gunnera* species, as assessed with scanning electron microscopy to define morphological characters of the exine and its structure to support its placement on the topology (*Wanntorp, Dettmann & Jarzen, 2004*).

## Biogeographic analyses

Distribution data were compiled from *Mora-Osejo, Pabón-Mora & González (2011)* and *Wanntorp & Wanntorp (2003)* together with records from the Global Biodiversity Information Facility (http://www.gbif.org) and regional herbaria (CONC and MEL) that were vetted by the authors. Using the extant distribution of *Gunnera* we defined nine operational areas for ancestral area estimation (Fig. 2): (A) northern Andes, from Venezuela and Colombia to northernmost Peru, corresponding to the páramos north of the Huancabamba Depression; (B) central Andes, from northern Peru (south of the Huancabamba Depression) south to the Tropic of Capricorn and including the Altiplano, Jalca, and Puna; (C) southern Andes, from northern Chile south to Patagonia, including the islands off the coasts of Chile and Argentina; (D) southeastern South America, including the lowlands of southeastern Brazil and the Rio Paraná drainage; (E) Mesoamerica, from southern Mexico to southernmost Panama; (F) the Hawaiian islands; (G) Africa, including Madagascar; (H) the Malay archipelago, including New Guinea; (I) Tasmania and New Zealand.

We inferred ancestral biogeographic ranges using the package BioGeoBEARS 0.2.1 (*Matzke, 2014*) in the R platform (*R Development Core Team, 2010*). BioGeoBEARS implements widely used models of range evolution (e.g., *Ree & Smith, 2008*), but it includes an additional parameter of cladogenetic speciation mediated by founder events: the jump parameter "j". This parameter allows daughter species to instantaneously "jump" outside the geographical range of parental species. We considered this model appropriate since several *Gunnera* species occur on oceanic islands (e.g., the Hawaiian and Juan Fernandez Islands). We inferred ancestral ranges across the *Gunnera* phylogeny using the Dispersal Extinction Cladogenesis (DEC) model with the J parameter (+j). The among-area connectivity was constrained in the following time slices as follows: northern and central Andean co-distributions were not permitted before 40 Ma (*Garzione et al., 2008*; *Hoorn et al., 2010*), Hawaiian distributions were not permitted before 30 Ma (*Clague et al., 2010*), lower connectivity (0.1 rate of dispersal) was set between Africa and South America throughout the last 90 Ma, as well as between South America and New Zealand plus Tasmania throughout the last 30 Ma (*McLoughlin, 2001*) (Table S2).

## Diversification rate analysis

To test for diversification rate shifts we used the software BAMM 2.0 (Bayesian Analysis of Macroevolutionary Mixtures; *Rabosky, 2014*). BAMM implements a Bayesian framework to estimate evolutionary rate parameters and explore candidate models of lineage diversification to quantify and detect heterogeneity in evolutionary rates. We ran BAMM using default priors for 1,000,000 generations sampling every 20 steps and accounting for incomplete taxon sampling. We analysed the output in R using the BAMMtools package 2.0.2 (*Rabosky, 2014*). We discarded the first 25% estimates as burn-in based on

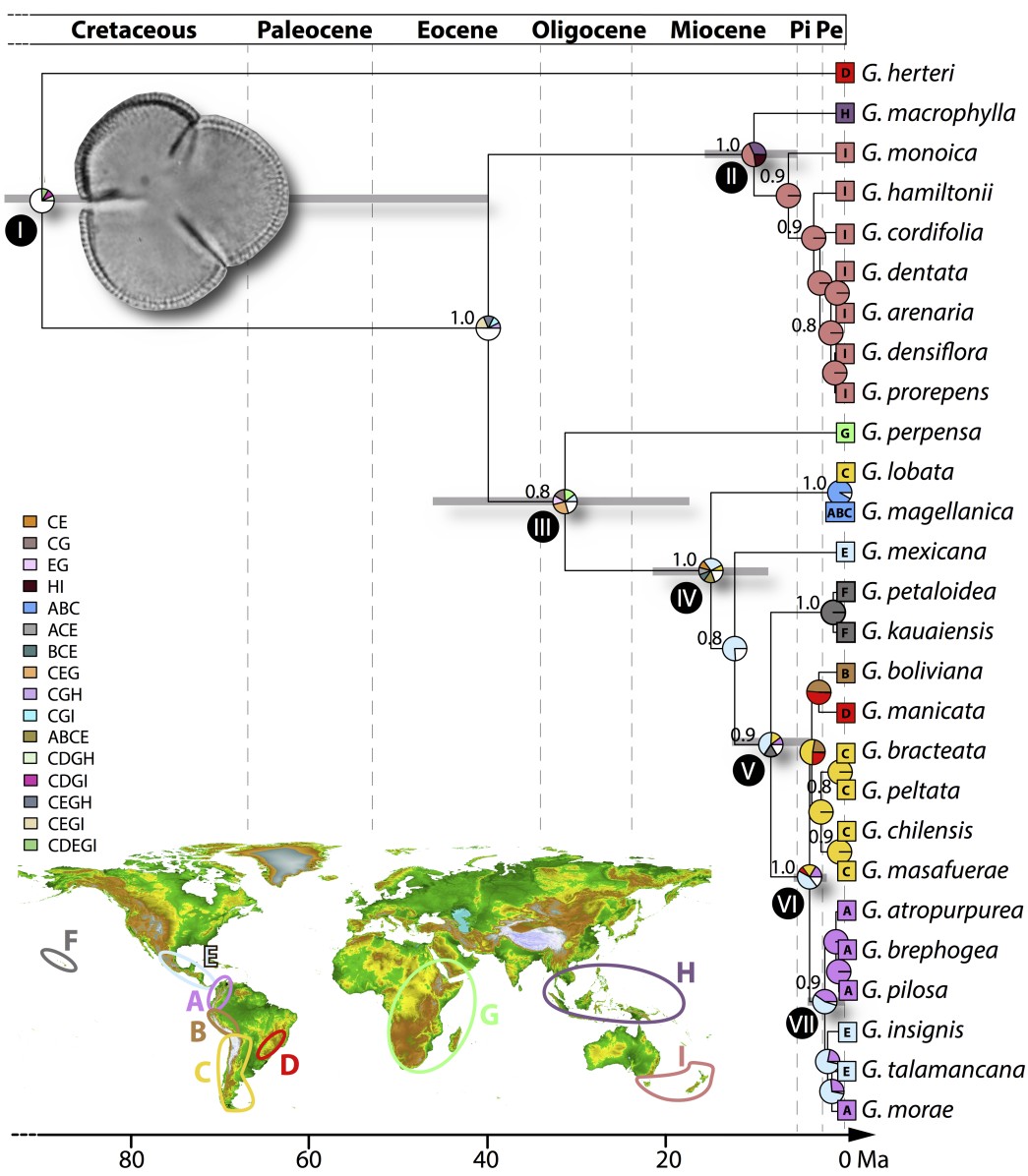

**Figure 2** **Biogeographic history based on the BioGeoBEARS optimization of the *Gunnera* topology calibrated in absolute time using *Tricolpites reticulatus* pollen.** Coloured boxes at the terminals of the phylogeny show the extant geographic distribution of species, with the colour legend defining the areas. All nodes over 0.8 PP are marked and node bars representing the 95% HPD time interval are shown for nodes of interest. Clades discussed in the text are marked with numerals I–VII. Uncertainty in ancestral range is shown where distributions with <5% probability of occurrence are combined into white portions in pie charts. Inset: Operational areas used: (A) northern Andes; (B) Central Andes; (C) southern Andes; (D) southeastern South America; (E) Central America; (F) Hawaii; (G) Africa; (H) the Malay Archipelago; (I) Tasmania and New Zealand; and other biogeographic areas based on combinations of those defined a priori.

the convergence of our data (effective sampling size of parameters greater than 200). We generated plots of net diversification and speciation rates through time and inferred the occurrence and position of rate shifts accounting for the 95% HPD of BAMM-inferred rate shifts based on a Bayes factor cut-off of 5.

## Climatic niche evolution

We performed ecological niche modeling for all 27 species of *Gunnera* included in the phylogenetic analysis using Maxent (*Phillips, Anderson & Schapire, 2006*) with eight WorldClim bioclimatic variables at a resolution of approximately 1 km$^2$ (*Hijmans et al., 2005*), following the methodology of *Evans et al. (2009)*. The bioclimatic variables associated with temperature were: mean annual temperature (MAT), minimum temperature of coldest month (MTCM), mean temperature of warmest quarter (MTWQ), and mean temperature of coldest quarter (MTCQ). Bioclimatic variables associated with precipitation were: mean annual precipitation (MAP), precipitation of wettest quarter (PWETQ), precipitation of driest quarter (PDQ), and precipitation of warmest quarter (PWARMQ). Climatic variables were chosen because of their biological meaning and by removal of the other 11 correlated WorldClim variables after a principal components analysis. We used a total of 882 vetted species occurrence points (see (b) above). From the ecological niche models we obtained the mean value for each of the eight bioclimatic variables weighted by the cumulative probability of each value (weighted mean; $_W$mean) to be used in comparative analyses, using the R package phyloclim (*Heibl, 2011*). To infer the climatic history of *Gunnera*, we used the projection of our phylogenetic tree in environmental (each bioclimatic variable) and temporal space assuming Brownian motion evolution (BM; *Evans et al., 2009*; *Schluter et al., 1997*), using the R package phytools (*Revell, 2012*).

Phylogenetic niche conservatism (PNC), defined as the retention of ecological traits, each bioclimatic variable ($_W$mean values), over time among related species (*Wiens et al., 2010*), was estimated using Pagel's lambda (*Pagel, 1994*) in the R package GEIGER (*Schluter et al., 1997*). Lambda ranges from one when trait evolution is strongly influenced by phylogeny, and a species niche-to-phylogeny correlation is equal to the Brownian model expectation, to zero when trait evolution is independent of phylogeny (*Pagel, 1999*). We used a likelihood ratio test (*Neyman & Pearson, 1928*) to determine whether the observed values differed significantly from zero and one. In general, phylogenetic signal indicates a statistical non-independence among species trait values due to relatedness, consistent with PNC (*Wiens et al., 2010*). To examine PNC explicitly we used the Akaike Information Criterion (*w*AIC) to compare the relative fit of three models of evolution to each bioclimatic variable ($_W$mean values). The models include: (i) a Brownian motion (BM) model of gradual and continuous drift; (ii) a stabilizing selection Ornstein–Uhlenbeck (OU) model with one optimum; and (iii) a white noise (WN) model of random variation, in which the similarity of species is independent of their phylogenetic relationships (*Hansen, Pienaar & Orzack, 2008*). We performed this comparison using GEIGER. The phylogenetic dependence of the realized climatic variation between related species, combined with the comparison of BM, OU, and WN models, provides an assessment for testing PNC (*Losos, 2008*; *Wiens et al., 2010*) and was thus suitable for our study.

## RESULTS

### Multispecies coalescent tree

The multispecies coalescent tree reconstructed here is consistent with the topology of the maximum parsimony tree based on molecular and morphological data of *Wanntorp & Wanntorp (2003)*. All major clades were resolved with moderate to high support (>0.80 posterior probabilities; Fig. 2), but some recent species-level relationships received poor branch support.

Wide time intervals (95% highest posterior density, HPD, values) were inferred for early nodes on the phylogeny (Nodes I and III, Fig. 2) but more recent nodes had little variation around the mean inferred age (e.g., Nodes VI and VII, Fig. 2). Overall the crown node of *Gunnera* was inferred at a mean age of 90 Ma (95% HPD 165–40 Ma). Following the origin and diversification of the genus in the Late Cretaceous, the two major clades of *Gunnera* originated between 31 (46–17 Ma, Node III, Fig. 2) and 10 Ma (15–5 Ma, Node II, Fig. 2). Contrasting results were resolved with regard to the time of origin of species, where *Gunnera herteri* had a mean age of 90 Ma and many others, particularly the Andean species, are younger than 2 Ma (Fig. 2)

### Biogeographic and diversification history

The BioGeoBEARS analysis resulted in broadly distributed ancestral nodes at the backbone of the tree, reflected in the uncertainty in ancestral range (distributions with <5% probability of occurrence combined are the white portions in pie charts; Fig. 2). Despite this, internal nodes ca. 12 Ma and younger were inferred with less ambiguity in ancestral area. These results reveal an ancient lineage that began extending its distribution significantly as of the Oligocene (ca. 30 Ma; node III; Fig. 2). BioGeoBEARS results show ambiguity in the biogeographic origin of the genus (node I; Fig. 2), but early lineages were distributed in the Southern Hemisphere (areas C, G, I).

Results from the BAMM diversification rate through time analysis shows that diversification in *Gunnera* generally increased over the last ca. 20 Ma (Fig. 3). However, this increase is almost exclusively due to the Pliocene diversification of the *Panke* clade, when the genus colonized the Andes. This is shown by a significant increase in net diversification rate in the *Panke* clade, either at its crown (Node VI, probability of 25%) or at its stem (Node V, probability of 22%). The highest rates of diversification were found during the Pleistocene in the *Panke* clade for the lineage distributed in the northern Andes.

### Climatic niches and history

Extant species of *Gunnera* are inferred to occupy cool (microthermal) and moderate (mesothermal) climatic conditions according to the classification of *Nix (1991)*, with a $_w$mean for mean annual temperature (MAT) ranging from 6.4° –20.3 °C and a $_w$mean mean annual precipitation (MAP) ranging from 813–3,588 mm (Fig. S2; Table S2). Microthermal species are distributed mainly in temperate latitudes of the Southern Hemisphere, with the exception of *G. talamancana*, which is found at high altitudes (1,900–3,400 m) in Central America (*Mora-Osejo, Pabón-Mora & González, 2011*). Mesothermal species are distributed in tropical latitudes in South and Central America, Hawaii, New

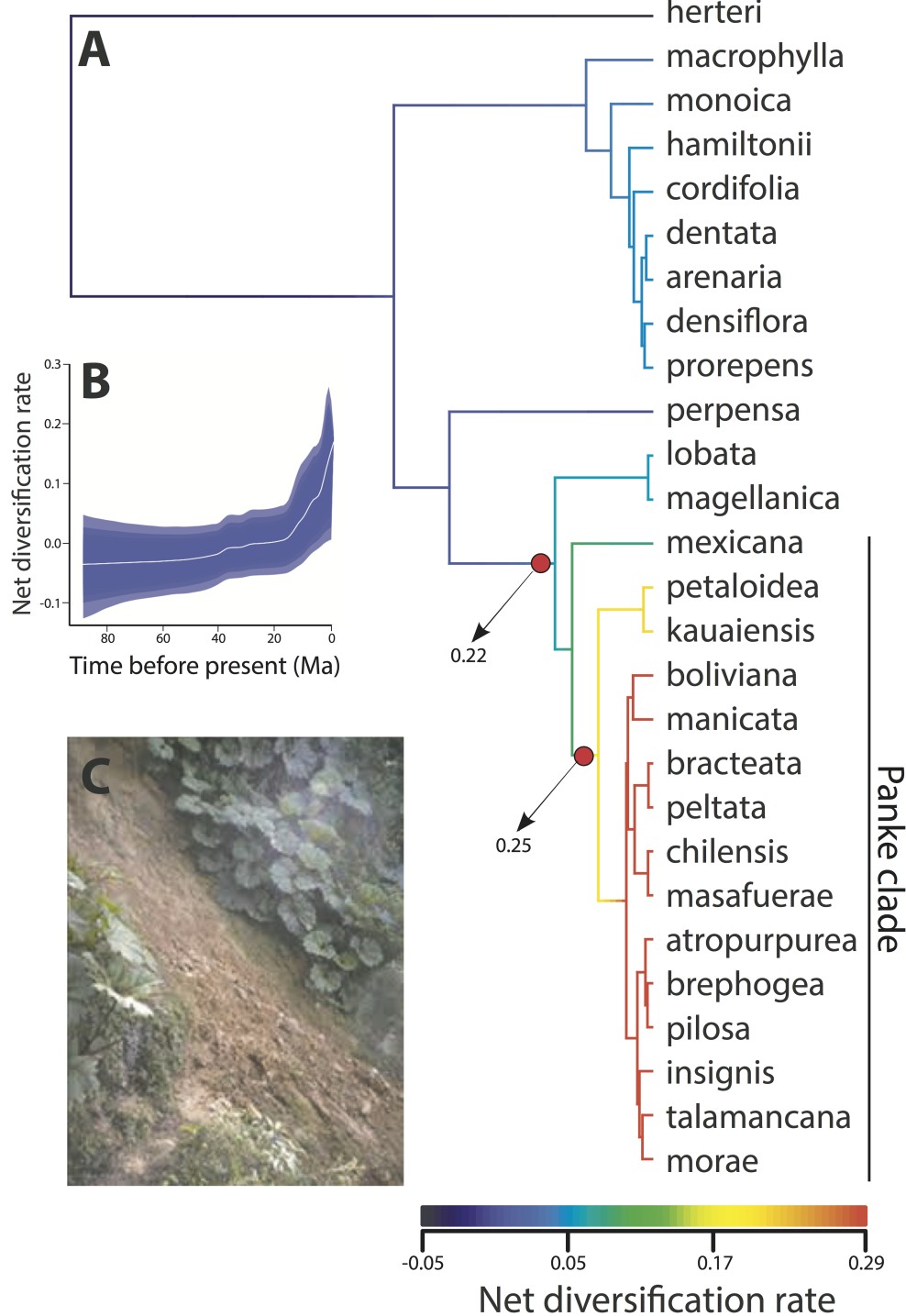

**Figure 3** **Diversification rate through time analysis using BAMM for all species sampled within _Gunnera._** (A) The results show a single, positive diversification rate shift, either at the stem (with 0.22 PP) or the crown (with 0.25 PP) node of the Andean Panke clade. (B) Results also support an increase in net diversification rate through time. (C) Some _Gunnera_ species are aggressive colonisers, here showing successful colonisation and persistence in the margins of a landslide in Costa Rica (image from Wikicommons).

Guinea, and Africa, with the exception of *G. arenaria* that occurs in temperate areas in New Zealand.

According to the estimation of ancestral climatic variables, the most recent common ancestor (MRCA) of *Gunnera* lived under a mesothermal climate *sensu Nix (1991)*, with a MAT of 15.3 °C and MAP of 1,577 mm. Phylogenetic signal using Pagel λ was detected for MAT ($\lambda = 0.93$) and mean temperature of coldest quarter (BIO11; $\lambda = 0.92$) and precipitation of wettest quarter (BIO16; $\lambda = 1$) and precipitation of warmest quarter (BIO18; $\lambda = 0.93$; Table S3). *w* AIC analyses showed that the evolution of the climatic niche is best described by the OU model, suggesting that selection pulls the climatic values toward an optimum. Minimum temperature of coldest month (Bio 6) and mean annual precipitation (Bio 12) showed no difference from a white noise model of evolution, indicating that these variables are independent of phylogenetic relationships in *Gunnera* species (Table S3).

## DISCUSSION

Based on the divergence times and relationships of the *Gunnera* multispecies coalescent tree, we examined the contribution of a southern hemisphere taxon to the mega-diverse tropical Andean flora of South America. Our results show the tempo of range expansion and lineage diversification.

### Biogeographic history of *Gunnera*

A question that has long intrigued biogeographers about widespread southern hemisphere lineages such as *Gunnera* is whether present-day disjunctions are the result of vicariance or dispersal events. Given the geographically extensive fossil record dating to the Cretaceous (*Jarzen, 1980*; *Wanntorp, Dettmann & Jarzen, 2004*) and the current distribution in all southern continents except Antarctica (*Bader, 1961*), *Gunnera* has long been considered a typical Gondwanan element (*Mora-Osejo, Pabón-Mora & González, 2011*; *Fuller & Hickey, 2005*), with vicariance proposed as the main driver of its current geographic distribution (*Wanntorp & Wanntorp, 2003*). Here we do not find support for unequivocal vicariance events in the biogeographical history of *Gunnera*, but instead interpret a general pattern of long distance dispersal from our results.

The mean crown age of 90 Ma for *Gunnera* is consistent with previous findings and the variation around the mean age (95% HPD 165–40 Ma) reflects what has been found in earlier work (*Bell, Soltis & Soltis, 2010*; *Magallón, et al, 2015*; *Tank et al., 2015*). *Gunnera* is inferred to have been widespread in the former Gondwanan territories including the southern Andes (area C), southeastern South America (area D), and Africa (area G at Node I in Fig. 2) during the Cretaceous. A Gondwanan distribution is also supported by several fossils from southernmost South America, southwest Africa, the Antarctic Peninsula, Australia, and Tasmania (*Macphail, 2007*). *Gunnera herteri* from South America is sister to the remaining *Gunnera* species, in agreement with previous studies (*Wanntorp & Wanntorp, 2003*; *Mora-Osejo, Pabón-Mora & González, 2011*; *Fuller & Hickey, 2005*), which supports a long history in the southern portions of the continent.

A vicariance event could be interpreted at the node where the Australasian lineages (Node II) diverge from the South American and African lineages (Node III), based on the phylogenetic pattern. However, the divergence time for that event dates to the Late Eocene (ca. 40 Ma) and geological evidence does not support division of these continents at that time (*Sanmartín & Ronquist, 2004*). Suitable areas for *Gunnera* in the tropical latitudes of New Guinea were available when this region emerged above sea level and Australia reached its current latitudinal position, both of which occurred at the end of the Cenozoic (12 Ma; *Fuller & Hickey, 2005*). In agreement with and according to our estimations the tropical lineage *G. macrophylla*, distributed from the Philippines and Sumatra east to New Guinea and the Solomon Islands, split from the southern Australia and Tasmania lineages at ca. 10 Ma. This result suggests that the arrival of extant *Gunnera* in tropical regions (e.g., area H) occurred via long distance dispersal, as has been inferred for other Gondwanan taxa such as *Nothofagus* (e.g., *Hinojosa et al., 2016*).

Another long distance dispersal is inferred at Node III where the African lineage *G. perpensa* split from rest of the genus at ca. 30 Ma, at the time when Africa and South America were completely separated (*Sanmartín & Ronquist, 2004*). Node IV joins the Andean species (areas A, B, C) with subgenus Panke (areas A, B, C, D, E, F). Node V infers a colonisation event to Hawaii (area F) and Node VI shows a division between southern South American species (areas B, C, and D) and those from the north of South America and Central America (areas A and E respectively).

## Range expansion in the Andes

*Gunnera* is inferred to have been present in southern South America since the Paleogene (Fig. 2). By the Mid-Miocene the stem node of the *Panke* clade (15 Ma; Node IV) was distributed in Central America and/or the Andes and began to diversify, both north into Central America and south again into the southern Andes (Fig. 2). The estimated mean age overlaps with that proposed for the formation of the Isthmus of Panama (*Montes et al., 2015*), a primarily terrestrial lowland region that has connected North and South America since ca. 15 Ma. *Bacon et al. (2015a)* and *Bacon et al. (2015b)* proposed that closure of the isthmus enabled taxa to expand their distributions both north and southwards during pulses of migration (ca. 23, 8, and 5 Ma). Range expansion at Node IV occurred after a long stasis (ca. 15 Ma) where there was a dearth of speciation or substantial extinction—two alternative explanations that are generally difficult to distinguish (*Crisp & Cook, 2009*).

The colonisation of the northern Andes (area A at Node VII, Fig. 2) is inferred to have likely taken place from Central American ancestors, sometime in the Early Pleistocene (ca. 2.27 Ma). Although the Andes began to rise as early as the Early Paleogene (*Garzione et al., 2008*), it is often difficult to disentangle the roles of Andean uplift and climate change on Neotropical diversification (but see *Lagomarsino et al., 2016*) because they occurred contemporaneously (*Gentry, 1982*). *Gunnera* is primarily a wet montane, rather than páramo, lineage, and does not require high elevations for successful dispersal.

## North American fossils: crown or stem relatives of *Panke*?

A sister relationship between the northernmost species of *Gunnera* (*G. mexicana*) and all other species in subgenus *Panke,* combined with the fact that there are several North

American fossil localities from the Late Cretaceous to the Eocene (80–50 Ma; *Jarzen, 1980*), led *Wanntorp & Wanntorp (2003)* to suggest that the South American species of *Panke* represented a recolonisation of South America from the north. This result is further supported by morphological similarities of leaf impressions and pollen size between the Late Cretaceous fossils and modern *Panke* species (*Jarzen, 1980*; *Wilkinson, 2000*), and the placement of the Hawaiian species *G. petaloidea* and *G. kauaiensis* as the next branching lineage after *G. mexicana.*

This scenario implies that the North American fossils belong to the crown or stem group of *Panke,* i.e., they would have been derived either from the branch connecting *Misandra* (*G. lobata* and *G. magellanica*) to *Panke* (Node IV, Fig. 2) or from the branch leading to *G. mexicana*. Although this is a possible conclusion based on the topology of our *Gunnera* phylogeny, the divergence times estimated here suggest otherwise. The splits connecting *Misandra* to the MRCA of *Panke* (where the dispersal to North America would have taken place) are estimated at ca. 15 Ma, which is almost 65 Ma later than the first documented North American fossils. Our results suggest that the North American fossils do not belong to the crown group of *Panke*. Instead, we suggest they likely represent a lineage that reached North America during the Cretaceous, but did not leave any living descendants. A similar scenario was found in the inconsistency between DNA-based divergence times and pollen fossils of *Nothofagus*, where 'incongruent' fossils might have been erroneously assigned to crown *Nothofagus,* whereas they in fact represented extinct stem relatives (*Hinojosa et al., 2016*; *Cook & Crisp, 2005*).

## Stasis versus rapid speciation

A remarkable aspect of our results is the striking difference in the stem ages of *Gunnera* species. *Gunnera herteri* is inferred to have originated 90 Ma during the Late Cretaceous (Node I), whereas 18 species in the phylogeny are inferred to be younger than 2 Ma (Fig. 2). The contrast between stasis and rapid speciation is also seen in the BAMM results (Fig. 3), where low net diversification rates are shown at ancestral branches and a shift in diversification rate detected in the Andean *Panke* clade is followed by a steady rate increase (Fig. 3C).

It is puzzling why some lineages have experienced long evolutionary stasis, whereas others underwent rapid speciation—all within the genetic constraints of a single clade. Our interpretation of stasis relies on an accurate inference of the dated tree, and wide credible intervals were detected for some nodes. Further, this result could be an artefact of extinction, if the lineages leading to the ancient species in fact diversified but all lineages except one went extinct (*Crisp & Cook, 2009*; *Antonelli & Sanmartín, 2011a*). However, there is palaeontological support for these exceptionally old stem ages. Fossil pollen on the Vega Peninsula of Antarctica dated from the Campanian/Late Maastrichtian have a distinctive exine that is nearly indistinguishable from that found in pollen grains of extant species of Australia, New Zealand, and Southeast Asia (*Wanntorp, Dettmann & Jarzen, 2004*).

The persistence of *Gunnera* for a longer time (ca. 90 Ma) than most other angiosperm genera is remarkable. We interpret our results with caution, but consider them as indicative of strong niche conservatism across the highly disjunct range of *Gunnera* for wet, montane

forest environments. Indeed, our climatic reconstruction shows a mesothermal niche preference for the crown node of *Gunnera* (Fig. S2), similar to that identified in other Cretaceous lineages (*DeConto et al., 1999*). Further, our estimate of phylogenetic signal shows high values (Table S3), particularly those associated with both mean annual and coldest quarter temperature variables, and an Ornstein–Uhlenbeck model, which together indicate that the climatic history of *Gunnera* underwent selection pressure (e.g., stabilizing selection) that favoured the ancestral niche over time (*Wiens et al., 2010*). Recently, *Hinojosa et al. (2016)* suggested that lineages of Gondwanan origin expanded into the tropics by tracking mesothermal climates. Dispersal towards current tropical zones has been possible because species have tracked ancestral climatic niches from high or mid-latitudes into lower latitudes, sometimes facilitated by climatic and geological changes.

It is interesting that *Gunnera*, a montane forest clade of Gondwanan origin, successfully dispersed to the northern tropical Andes, where few other similar plant clades could. A key element to this may be due to its colonising nature. *Gunnera* comprises species of forest edges and marginal habitats (*Greimler et al., 2013*), appear after landslides in wet forests (Fig. 3C; *Vanacker et al., 2003*), underwent long distance dispersal events to Hawaiian and the Juan Fernandez Islands, and is persistent in the seed bank (*Arroyo, Cavieres & Humana, 2004*; *Fesq-Martin et al., 2004*). *Gunnera* species also quickly colonise glacial forelands (*Perez et al., 2014*) and their pollen is commonly found in glacial and post-glacial sediments in both southern South America and in Tasmania (*Heusser, Heusser & Hauser, 1992*; *McKensie & Kershaw, 2000*). Lastly and potentially most convincingly, the fact that some *Gunnera* species are invasive (*Fennell et al., 2013*; *Skeffington & Hall, 2011*) clearly shows their aggressive colonising abilities that likely differentiate them from other montane plant groups.

## Comparison with other Andean groups

There are multiple examples of plant clades that have colonised the Andes from the north, as we suggest for the Panke clade of *Gunnera*. Some are 'boreotropical' elements that probably reached South America around the Palaeocene-Eocene Thermal Maximum (∼55 Ma; *Zachos, Dickens & Zeebe, 2008*), when a large belt of tropical forest covered much of southern Laurasia, thus functioning as a biotic corridor for Palaeotropical lineages (e.g., *Bacon, Baker & Simmons, 2012*; *Antonelli et al., 2009*). Later, dispersals southwards from North to South America may have been facilitated by the Greater Antilles and the Aves Ridge around the Eocene/Oligocene boundary (*Iturralde-Vinent & MacPhee, 1999*), and finally through the Panama Isthmus after its uplift ca. 15 Ma (*Montes et al., 2015*). Examples of northern taxa colonising the Andes with local radiations include *Hedyosmum* (*Antonelli & Sanmartín, 2011a*; *Zhang, Feild & Antonelli, 2015*) and *Lupinus* (*Hughes & Eastwood, 2006*). Many of these northern lineages have undergone significant radiation in the páramo.

In contrast, there is relatively little evidence of Andean plant clades that are derived from the south. Some well-known southern Hemisphere ("Gondwana") groups, such as *Nothofagus* and *Araucaria*, simply do not enter the Andean tropical zone, but reach tropical areas in Australasia (*Iturralde-Vinent & MacPhee, 1999*). *Fuchsia*, which is considered to

have a southern origin based on a rich Antarctic Cenozoic fossil record, does not show a clear biogeographic pattern from molecular phylogenies (*Berry et al., 2004*). One robust example is the Andean wax palms, that have moved up the Andean cordilleras from the central portion of the range northwards in a stepping stone fashion (*Ceroxylon*, see *Sanín et al., 2016* for a review of the pattern).

## CONCLUSIONS

For many decades *Gunnera* has attracted the attention of botanists and biogeographers concerned with southern hemisphere disjunctions and the break-up of Gondwana. Here we have shown that it also constitutes a model taxon to study biogeography in general, as well as the colonisation and diversification of southern elements in the tropical Andes in particular. The extraordinary species longevity inferred here for species in Southeast Asia, Africa and eastern Brazil—ca. 90 Ma according to our results—is most likely due to climatic, and potentially morphological, conservatism despite the massive geotectonic and climatic changes that took place during its history. In contrast, the recent and rapid diversification of Andean lineages could be explained by the massive increase in the area of suitable habitats and opportunities for allopatric speciation, as a consequence of the Andean uplift and late Neogene climatic changes. Understanding the evolution of Andean mega-diversity thus requires identifying and tracing the diversification of southern, northern and locally derived taxa.

## ACKNOWLEDGEMENTS

We would like to thank C Hughes and U Swenson for reading and commenting on early versions of the manuscript and S Razafimandimbison for attempts to sequence ITS for *Myrothamnus*. We are also thankful to Herbarium GB for allowing us to extract samples for DNA analyses. Vivian Aldén provided laboratory assistance. The analyses were performed on the bioinformatics computer cluster Albiorix at the Department of Biological and Environmental Sciences, University of Gothenburg.

### Funding

Financial support was provided by the Swedish Research Council (B0569601), the European Research Council under the European Union's Seventh Framework Programme (FP/2007-2013, ERC Grant Agreement n. 331024), the Swedish Foundation for Strategic Research and the Knut and Alice Wallenberg Foundation (through a Wallenberg Academy Fellowship) to Alexandre Antonelli. Luis F. Hinojosa was funded by FONDECYT 1150690, Millennium Institute of Ecology and Biodiversity (IEB) Grant P05-002 from MIDEPLAN (Chile), PFB-023 from CONICYT (Chile). The funders had no role in study design, data collection and analysis, decision to publish, or preparation of the manuscript.

## Grant Disclosures

The following grant information was disclosed by the authors:
Swedish Research Council: B0569601.
European Research Council: 331024.
Swedish Foundation for Strategic Research.
Knut and Alice Wallenberg Foundation.
FONDECYT: 1150690.
Millennium Institute of Ecology and Biodiversity (IEB): P05-002.
CONICYT: PFB-023.

## Competing Interests

The authors declare there are no competing interests.

## Author Contributions

- Christine D. Bacon conceived and designed the experiments, analyzed the data, prepared figures and/or tables, authored or reviewed drafts of the paper.
- Francisco J. Velásquez-Puentes and Luis Felipe Hinojosa performed the experiments, analyzed the data, prepared figures and/or tables, authored or reviewed drafts of the paper.
- Thomas Schwartz performed the experiments, authored or reviewed drafts of the paper.
- Bengt Oxelman and Bernard Pfeil analyzed the data, authored or reviewed drafts of the paper.
- Mary T.K. Arroyo authored or reviewed drafts of the paper.
- Livia Wanntorp performed the experiments, contributed reagents/materials/analysis tools, authored or reviewed drafts of the paper.
- Alexandre Antonelli conceived and designed the experiments, contributed reagents/materials/analysis tools, authored or reviewed drafts of the paper.

## DNA Deposition

The following information was supplied regarding the deposition of DNA sequences:
DNA sequences have been deposited in GenBank. A list of the accession numbers can be found in File S1.

## Data Availability

Raw data is provided in the Supplemental Information.

## Supplemental Information

Supplemental information for this article can be found online at http://dx.doi.org/10.7717/peerj.4388#supplemental-information.

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
