# Peer review of "Evolutionary persistence in Gunnera and the contribution of southern plant groups to the tropical Andes biodiversity hotspot"

_PeerJ, doi:10.7717/peerj.4388_

## Round 0.1 · original submission · Minor Revisions

The reviewers found the paper interesting and relevant and recommend publication pending minor revisions.

Nevertheless, the reviewers made very important suggestions and raise relevant points that you should be considered and improved in the next version of the manuscript. Please evaluate all the different comments of both reviewers and correct and answer them one by one for the next submission of you study.

Reviewer 1 makes important methodological questions that should be carefully answered. Especially answer in detail the questions about the dates and fossil and the points the Reviewer asks about the programs and methods used in different analyses.

Reviewer 2 has also some methodological questions, and in particular asks for a better explanation regarding the DNA sequences: if they are new or not (and then where they were originally published) and about the information about their Genbank numbers, collecting places, etc. that are very important.

This reviewer also asks you to include Figures S2 and Table S1 in the main text, not in supplementary information. I agree with this suggestion, and would also suggest to include Figure S1 in the main text, as this will make the paper more interesting and attractive to the readers.

Reviewer 3 also has relevant methodological comments in relation to the dates, points of calibration and the BAMM and BBM methods, and other comments regarding hots spots of diversity and the figures.

I consider that Figure 2 can be improved: I feel that the photo of the pollen grain is too big (and it is not too obvious what it is, and for instance, no other pollen grain is shown). Also explain the meaning of the colors squares and letters in the left of the figure.

Reviewer 1 ·

Basic reporting

Overall, the manuscripts is well executed.

Experimental design

Some of the methods need more details and clearer justification.

Validity of the findings

The findings are interesting and robust (although there are some concerns).

Additional comments

The paper addresses a significant question on plant diversity in the Neotropics, that of the history of diversification of plants in the mega-diverse Andean hotspot. The manuscript is well written, with a good structure and flow of ideas. The objectives are quite clear and the methods are appropriate for the questions being asked. The methods seem reasonable and overall robust, although I have some concerns in this area (see comments below). The discussion is adequate, but in need of careful revision. In addition, some of the interpretations of the results are, in my opinion, a bit to far-fetched; specially the section of stasis vs. rapid speciation.
Finally, I would like to see a more integrated discussion on the history of Gunnera, as is I found the sections somehow loosely connected.
Nevertheless, I think the present manuscript would represent a significant contribution to the understanding of Andean (an other montane regions) mega-diversity and I would recommend its publication pending revisions.

My main concerns are related to some of the methods used and some of the interpretation in the discussion. The methodological parts do not appear to be critical, but I am a bit skeptical on several fronts. So some clarifications are needed and extra details on specific analyses would be welcomed. Several comments below.

Dating analyses:
I’m not very familiar with the STACEY algorithm, but it appears that its main strength comes when using many loci, for which the coalescent is the best way to go. My question goes to the fact that you are effectively using 3 loci (cpDNA + two nuclear genes), so what are the real advantages of using STACEY over other dating algorithms?

According to Wanntrop & Wanntrop, Tricolpites fossils show a great resemblance to specific extant groups of Gunnera, but the authors did not perform a proper phylogenetic analyses to place these fossils within the phylogeny. So I am a bit skeptical whether the fossil might belong to some stem member of Gunneraceae. From the original description of the fossil grains, these probably belong to crown Gunnera due to their resemblance to different pollen types within the family. Although this assignment might actually hold, since the dating is so crucial here I think more justification is needed on the placement of the fossil in crown group Gunneraceae.

I'm a bit concerned about your interpretation of the age of Gunnera because of the very wide interval you obtained. The mean age of 90 Mya for CG Gunneraceae is reasonable, but as said before I have doubts regarding this particular fossil calibration. What happens if the fossil goes to the stem group instead? Remember that the first unambiguous evidence of Core Eudictos is ca 100 Mya, so having CG Gunneraceae older than this is quite surprising. More so, Gunneraceae going back into the Early Cretaceous-Late Jurassic is at odds with everything we know about the fossil record of angiosperms: 165 will push Core Eudicots to a time prior to the first fossil angiosperms.

BAMM analyses:
More detailed is needed for the BAMM analyses and the corresponding results.
Moore et al. (2016) showed that the posterior distribution of range shift estimated in BAMM is highly influenced by the prior. How did your analyses behave in this respect? and more importantly, what prior on the number of shifts was used?
A major drawback of BAMM (as with other methods that estimate diversification rates) is that rates of speciation and extinction are hard to tease apart and thus interpretations based on these are limited. Authors should at least acknowledge this problem when discussing speciation dynamics.

Climatic niche evolution:
I’m a bit concerned on the way authors selected climatic variables. Did authors estimate the correlation matrix among variables for all species pooled together? If so, I’m not sure this works because the correlation matrix would be heterogeneous in different areas and for different species. I never been particularly concerned by correlated variables in MaxEnt (but I may be seriously wrong), so I would go with the 19 climatic variables (maybe eliminating the monthly variables only).
The cumulative output of Maxent seems reasonable, but a bit of explanation would be useful.
If I understood correctly, the climatic history of Gunnera was reconstructed under a BM model, but the tests of PNC indicate that this is best modeled by a OU model. Please clarify.
What would be the possible effect of missing species in the reconstruction (14 out of 41 for the Panke group)? I'm not concerned on BAMM estimates, but trait reconstruction is highly conditioned on the observed traits (sampled species). Are the missing species climatically similar to others?
Figure S2 is hard to follow. Maybe incorporating climatic niche information into figure 2 might help, or viceversa. Authors might have other ideas. By the way, why are only figures for temperature and precipitation shown in figure S2?

Results and discussion:
Lines 336-337. How do you reconcile the fact (from your own figure) that Gunnera fossils are found in the Andean region during the Cretaceous. Why did the genus not start to diversify earlier? Are you assuming these fossils do not belong to Panke?

Lines 346-348: So, Gunnera is effectively a northern migrant, rather than a southern group that colonized the Andes. This seems interesting. Please clarify, this goes strait to the main objectives and definitions of the manuscript.

Lines 359-361. No, this argument is very very tricky. By stating this you are implying a very high confidence on your selection of some fossils over others (besides doubts that can be raised regarding the assignment of those fossils to particular nodes in the phylogeny). Your dated phylogeny relies in assuming certain fossils are part of specific groups. So, if these fossils turn out to be part of Panke, your dated tree would be wrong. Hopefully, the point I'm trying to make here is clear.

Lines 362-363. But you do not know where this lineage goes in the phylogeny, which is crucial to support or reject your dating hypothesis. So, you have to be more careful here.

Lines 372-274. Low rates of diversification do not mean stasis or low speciation. Extinction is crucial here.

Lines 382-383. I do not see the connection.
First, the fossils have not been properly assigned to a particular branch in the Gunnera tree, it is just mere resemblance we are taking about here. Second, these fossils are evidence of extinct members of Gunnera, which might turn out to be part of the G. herteri clade. I think you are pushing your results to much. To me, the diversification results of BAMM are interesting enough, without going into speciation or extinction (which by the way are not shown anywhere). The two basal branches show very contrasting histories of diversification. The question here would be why the Panke lineage is diversifying more rapidly than the G. herteri lineage?Perhaps the climatic history can identify something here.

Lines 384-385. To me, the 165 Mya is a highly unreasonable age (it is misleading in figure 2). And if you are using the confidence interval to make a point, you can certainly also make the opposite point with the 40 Mya lower bound, which would make your results a lot less contrasting in terms of age. I think there is not much to say (confidently) about the age of G. herteri given the wide confidence interval, or just stick to the mean.

Lines 394-396. You have to keep in mind that you have older fossils in northern latitudes. I think this goes to the heart of another of my main concerns. I feel that the fossil record is somehow disconnected from the rest of the discussion. It would certainly be nice to include a more explicit account of the distribution of fossils throughout the discussion.

Lines 398-399. To me, the key element here is that Gunnera in the Andes comes from the north, not the south. So perhaps, the ability to colonize the Andes came from the first northward shift in the distribution of the genus.

Lines 424-427. To me, the pattern observed in Ceroxylon is quite different from yours, with a south to north (stepping-stone like) diversification. Whereas the pattern you uncovered in Gunnera is more complex, a south to north expansion, followed by a north to south diversification. Again, I think that explicitly integrating the data on the distribution of fossils would be highly valuable.

Reviewer 2 ·

Basic reporting

The authors present a case study (the genus Gunnera) to investigate the role of southern immigrants in the assembly of the Andean biodiversity hotspot flora. This is an interesting case study since it has been hypothesized to represent a Gondwanan origin with vicariance by continental drift to explain the present-day distribution of most Gunnera species. The authors present a sound analytical and interdisciplinary approach. The article is very enjoyable to read and is mostly well written. I have a few comments aimed at improving the clarity of the manuscript, providing more detail on the methodology that should allow someone to replicate the study, and some comments on the interpretation of the results and sentences that sound too speculative to me. If correctly addressed, in my opinion, the manuscript is suitable for publication in PeerJ.

The Abstract gives the impression that one of the main results of the paper is that Gunnera colonized the Andes or northern Andes from the south. Please re-rephrase the last sentence to be consistent with what you say in the discussion lines 409-410, colonization from the north (Central America). In the last sentence of the abstract, the authors say “composite origin”, does this refer to a northern and southern colonization of the Andes? I would argue that this is not an example of an Andean colonization from the south.
This is my main interpretation comment. From your figure 2, I see Andean colonization from the north (Central America) not the south (node IV and inner nodes). I provide a further explanation below (Discussion, Lines 327-328).

Introduction:

Line 71. “…below and above the treeline”. Do you mean north and south of the treeline?

Lines 73-75. Sentence needs revision since it is unclear. “Unlike the numerous examples of páramo species from the southern regions of the Andes colonized the northern Andes, fewer montane forest dwellers, such as Gunnera, have successfully dispersed to the region”. From which region to which region?

Lines 100-101. “We also discuss why so few southern hemisphere montane forest dwellers have entered the northern Andes….” The sentence sounds vague. Can you provide an idea of how many is few? What is the proportion of those southern hemisphere montane forest dwellers that have colonized the northern Andes? Just to get an idea of how rare this event is.

It is unclear whether there is a previous molecular phylogeny of Gunnera and how much DNA data or information was taken from a previous study.

Experimental design

Methods:

The authors should give a percentage of taxon sampling of Gunnera in the text. Total percent taxon sampling of the genus?

Are all DNA sequences newly generated for this study? How much was obtained from Genbank? I did not find a table with this information: Taxa names, voucher specimens, Genbank numbers, collecting place, etc.

Lines 136-138. “The substitution model was GTR with a gamma prior distribution (0.05, 10) on each substitution type…” Is this information the same as the information given after the comma, which describes another gamma distribution? For all three linkage units? Not loci.

Line 148. Mean age of 90 Ma. Is this all the information needed to replicate the dating analysis? Any other settings used? Type of prior distribution?

Line 173. Was the DEC +J model selected according to the AIC? BioGeoBEARS implements several other range evolution models. An explanation of why the DEC+J model was chosen is needed.

Lines 173-178. Time slices and rates/probabilities of dispersal between biogeographic areas are not clear or enough in the description provided. Readers shall be able to replicate your analysis; in its current state, this is not possible. I suggest to add a dispersal cost matrix table.

Line 196. “climatic variables were” , same for Line 199, replace with “were”

Line 200. Please revise this sentence, English is awkward. Replace with “We chose”, “because of their biological”. It is unclear how climatic variables were eliminated after the PCA. Did the authors take axes 1, 2 and 3 and selected the environmental variable with the highest load on each? What was the cutoff value of the load? How many axes did the authors choose?
Isn’t the purpose of a PCA to use the first 2 or 3 axes as the new environmental variables instead of the original Worldclim variables?

Line 203-205. This sentence needs revision. It is unclear why this test was conducted. Perhaps start the sentence with something like this “To infer the climatic history of Gunnera….”.
It seems that the mean values of each of the 8 bioclimatic variables for each Gunnera species was obtained after the Maxent models were obtained? A better description is needed and a better integration with the next sentence (Lines 206-208) since it seems that it is part of the same analysis.

Line 209. Unclear which variables were used for the PNC analysis. I am guessing the 8 bioclimatic variables.

Line 223 “combined with the comparison of BM and OU models…” but not with the white noise model?

Results:

Line 247-248. Which area(s) are represented by the white sections? Does it mean all areas or combination of areas with LESS than 5% probability?

Lines 273-275. Consider deleting “between temperature variables” and “between precipitation variables” in this sentence since it reads awkward. Also, the bioclimatic variable acronyms are not working because these are too long (e.g. MTCQ, PWETQ, PWARMQ). I would rather see them spelled out completely like the authors did on the last sentence of that paragraph “Minimum temperature of coldest month (Bio 6) and mean annual precipitation (Bio 12)”

Validity of the findings

Discussion:

Line 323. Another long distance event is from area E (Central America) to C (southern Andes) at node VI.

Lines 327-328. “By the Mid-Miocene the stem node of the Panke clade (15 Ma; Node IV) was distributed in the Andes…..”
Based on your Figure 2, node 4 (and subsequent inner nodes) had a higher probability for area E (Central America). We can thus infer that Central America was the ancestral area for the Andean Gunnera species. Can the authors provide an explanation on why they chose to interpret that this node IV represented an Andean-distributed ancestor?

Lines 361-366. I like the interpretation here. It shows how important morphological studies are to correctly place a fossil in a phylogenetic tree.

Lines 382-283. “This pollen evidence suggests a lack of extinction bias and hints to PNC as an important mechanism behind evolutionary stasis.” I don’t follow the logic or conclusion of this sentence. The fact that PNC may have existed for the exine (empirically untested) does not exclude extinction bias. How can PNC of one character exclude extinction bias? And how can PNC of the exine explain the evolutionary stasis? (given that there could be several other phenotypic characters not under PNC).

Lines 419-420. I would delete this part of the sentence “as our results show more generally for Gunnera”. This last part contradicts what the authors wrote in the previous paragraph (Lines 409-410). I agree with the interpretation of the biogeographic analysis that colonization of the Andean Gunnera species came from the north (Central America).

Lines 424-427. The last sentence in the Discussion sounds weak. “there are several taxa that are similar to Gunnera” Can the author name some? Isn’t the goal of this last paragraph to provide examples of lineages that colonized the Andes or northern Andes from the South? Ceroxylon colonized the northern Andes from the south. This is a different pattern than the one shown here for Gunnera.

Conclusions

Line 435. Delete “morphological” since no test of PNC was conducted on morphology.

Line 437-438. This sentence sounds too speculative. “best explained by the massive increase in the area of suitable habitats and opportunities for allopatric speciation…” Since there is no test presented to address allopatric speciation, I would replace with “could be explained” and delete “ and opportunities for allopatric speciation”.

Additional comments

Figure 2: What does the white part of the pie charts mean? What is that number 165 Ma in node I? An explanation should appear in the figure caption to allow the figure to stand-alone.

Figure 3: Add genus name (Gunnera) to each taxa in the phylogeny and italicize names throughout.

Supplementary figure S2 and Table S1 could go in the main text, not as supplementary. Since this is an open access journal, I don’t think there is a space limitation, is there? This figure and table disclose important results of the paper.

A few typos:
Line 67 “into”
In reference 15: “evolution”, “fire”. In reference 75: “Trachycarpeae”

Reviewer 3 ·

Basic reporting

Clear, unambiguous, professional English language used throughout.

Line 31: Note that the use of “hottest” is subjective. Please consider revising and use “…is the richest and most diverse--”. Please consider “ancestral area reconstruction (lines 89–90). In general, the text is clear, unambiguous, and professional English language is used throughout.

Intro & background to show context. Literature well referenced & relevant.

Structure conforms to PeerJ standards, discipline norm, or improved for clarity.

Figures are relevant, high quality, well labelled & described. Perhaps a better photograph of a Gunnera species is desirable; the one included is low quality and blurred.

Raw data supplied (see PeerJ policy). 26]. All new sequences generated in this study have been deposited in GenBank (Appendix 1). Though occurrence records should be publicly available, perhaps at Dryad. The records are illustrated in Fig. 1, but the resolution at the scale used is not very helpful.

Experimental design

Original primary research within Scope of the journal.
Research question well defined, relevant & meaningful. It is stated how the research fills an identified knowledge gap.
Rigorous investigation performed to a high technical & ethical standard.
Methods described with sufficient detail & information to replicate.

Lines 81–84. According to the authors, Gunnera had a wide geographic distribution during the Cretaceous, as demonstrated by numerous fossil pollen records from the Antarctic Peninsula, Australia, the Kerguelen Plateau, as well as in both North and South America according to Jarzen’s (1980) data of Gunnera pollen occurrence in the fossil record (lines 81–84). The oldest of these dates to the Turonian (ca. 90 Ma) of Peru [22] and ten million years later Gunnera became widespread across all landmasses that previously formed Gondwana (Jarzen 1980). However, they only used one date for tree calibration. Please explain why the other more recent dates of pollen fossil records shown in Fig. 1 are not useful for calibration.

Lines 144–151. Also, the use of pollen records may be problematic for divergence time estimation (lines 144–151). The shortcomings of fossil pollen use in divergence-dating analysis must be acknowledged. Fossil pollen often has very high representation, especially for wind-pollinated species, and therefore, pollen produced by members of a given clade can appear in the fossil record relatively soon after the origin of that clade. However, fossil pollen is typically identified with relatively low taxonomic resolution, a problem exacerbated by the fact that the identification of fossil pollen types is rarely supported by synapomorphies and is often based solely on gross similarity. As a result, extinct plesiomorphic types can be confused with extant groups. This problem can be alleviated when fossil pollen data are accompanied with macrofossils or mesofossils (i.e., leaves, flowers and fruits). Please indicate whether or not these concerns apply to your study.

Lines 144–151. An alternative for tree calibration is to increase taxon sampling as to include secondary calibrations from tree-calibrated phylogenies that used many fossil calibrations. For example, Magallón et al. (2015) reports a split between Gunneraceae and Myrothamnaceae dated at 104.6 MA and the split of Gunnerales at 129 Ma.

Lines 108–110. It is unclear whether you have used outgroups for estimation of phylogenetic relationships and divergence timing. Please clarify this and indicate the taxa that were used as outgroups of Gunnera.

Lines 166–178. Please consider comparing your results of the dispersal–extinction-cladogenesis model (J-DEC) for a likelihood framework with those using a Bayesian Binary MCMC analysis (BBM). The BBM analysis takes into account the inherent uncertainty of the phylogenetic inference by optimizing ancestral areas over multiple trees (Nylander et al., 2008).

Lines 181–190. From my reading, it is not clear whether you have estimated both speciation and extinction rates for phylogenies simulated under constant-rate birth-death process in BAMM. BAMM (Bayesian Analysis of Macro-evolutionary Mixtures) has become widely used for estimating diversification rates and rate shifts. At the same time, several papers have concluded that estimates of net diversification rates from the method-of-moments (MS) estimators are inaccurate. Meyer & Wiens (2017) have recently compared the ability of these two methods to accurately estimate clade diversification rates. They uses simulations to compare their performance, and found that BAMM yielded relatively weak relationships between true and estimated diversification rates. This occurred because BAMM underestimated the number of rates shifts across each tree, and assigned high rates to small clades with low rates. Errors in both speciation and extinction rates contributed to these errors, showing that using BAMM to estimate only speciation rates is also problematic. In contrast, the MS estimators (particularly using stem group ages), yielded stronger relationships between true and estimated diversification rates, by roughly twofold. Furthermore, the MS approach remained relatively accurate when diversification rates were heterogeneous within clades, despite the widespread assumption that it requires constant rates within clades. Overall, they caution that BAMM may be problematic for estimating diversification rates and rate shifts. Please discuss some of the caveats discussed by Meyer and Wiens (2017) Evolution DOI: 10.1111/evo.13378.

Validity of the findings

Their conclusions are well stated, linked to original research question and limited to supporting results. However, there are some important methodological issues indicated in the Experimental design section that must be corrected or clarified before this manuscript is recommended for publication in PeerJ.

Additional comments

I really enjoyed the reading of this manuscript. I have no major concerns regarding the quality of the study.

---

## Round 0.2 · Minor Revisions

I want to acknowledge the authors for answering the methodological and conceptual comments and the other suggestions of the three reviewers.

The new version is almost ready, but I have some minor issues.

1) Line 69 mentions that the subgenus Panke includes 41 species.
Line 97 in says that subgenus Panke has 40 species: one of the two numbers is not correct, please check the right number.
Also in lines 96 and 97, the numbers do not seem to add up: If the genus has 58 species, and you sequenced 27 species, 58-27= 31 species were not sequenced in total; the sequenced species included all species of 5 subgenera, thus apparently only missing species from the Panke subgenus.

But in the next line, you say that you sequenced 14 out of a total of 40 species in the Panke subgenus, 40-14= 26 species not sequenced.
Then there seem to be 5 not accounted species, either they belong to some other subgenus, or to Panke, but then either the numbers are wrong, or the explanations are not clear. Please check and correct.

2) Lines 358 and 405, to better reflect you findings, and to be conservative (for instance, according to Magallón et al. (2015; New Phytologist (2015) doi: 10.1111/nph.13264) the date of origin of the Angiosperms is between 136 and 139.35 Ma. with a 95% confidence interval making the time you mention, 165 Ma , older than the angiosperms), I would suggest to change:

In line 358: “The persistence of Gunnera for a longer time (ca. 90Ma) then most other…”

In line 405: “…and eastern Brazil –- ca. 90 Ma according…”.

Reviewer 1 ·

Basic reporting

no comment

Experimental design

no comment

Validity of the findings

no comment

Additional comments

This is my second time reviewing this manuscript and again found a very well executed analysis.
As for my original review, my concerns were dealt with appropriately in the new version of the manuscript and thus I have no further comments. The methods are clearer and better justified and I found a much more cohesive argument in the results and discussion.
The present manuscript is very interesting and addresses a relevant question for the understanding of biodiversity in the Neotropics. Thus I am happy to recommend its publication.

Reviewer 2 ·

Basic reporting

I am satisfied with the way the authors have addressed my comments.
I have no further suggestions and think the manuscript is ready for publication.

Experimental design

All good

Validity of the findings

All good

Additional comments

All good.

---

## Round 0.3 · accepted · Accept

I am very pleased with the final version of the manuscript.
I think this is a very interesting and relevant contribution to understanding the evolution of this fascinating genus and also to advance in disentangling the complex causes and patterns of the Neotropical hyper-diversity of angiosperms and other organisms.